

# Monitoring long-term changes of glacial seismic activity with continuous seismological observations: a case study from Spitsbergen

**W. Gajek[1*], J. Trojanowski[1], M. Malinowski[1]**

[1]{ Institute of Geophysics, Polish Academy of Sciences, Warsaw, Poland}

*corresponding author: wgajek@igf.edu.pl

**Abstract**

Changes in the global temperature balance have proved to have a major impact on the cryosphere and therefore retreating glaciers are the symbol of the warming climate. Long-term measurements of geophysical parameters provide the insight into the dynamics of those processes over many years. Here we explore the possibility of using data recorded by permanent seismological stations to monitor glacial seismic activity. Our study focuses on year-to-year changes in seismicity of the Hansbreen glacier (southern Spitsbergen). We have processed 7-year-long continuous seismological data recorded by a broadband station located in the fjord of Hornsund, obtaining seismicity distribution between 2008 and 2014. To distinguish between glacier- and non-glacier-origin events with the data from only one seismic station in the area, we developed a new fuzzy logic algorithm based on the seismic signal frequency and the energy flow analysis. Our research has revealed that the number of detected glacier-origin events over the last two years has doubled. We also observed that the annual events distribution correlates well with the temperature and precipitation data. In order to further support our observations, we have analysed 5-year-long seismological data recorded by a broadband station located in Ny-Ålesund (western Spitsbergen). Distribution of glacier-origin tremors detected in the vicinity of the Kronebreen glacier shows a steady increase from year to year, however not as significant as for the Hornsund dataset.

*keywords: icequake, ice-vibration, glacier, Spitsbergen, seismic monitoring, fuzzy logic*



## 1 Introduction

Glaciers are dynamic systems in constant motion. Their movement itself, changes of strain regime, cracks and crevasses opening, calving and ice-basal friction release energy, which can excite detectable seismic signals in a broad frequency range (from seismic to seismo-acoustic). Therefore, glaciers' seismicity was studied through many years with growing interest.

Pioneering studies in the 60s and 70s characterised icequakes parameters and related seismicity to the ice movement and internal ice stress changes (Lewandowska and Teisseyre, 1964; Neave and Savage, 1970; Cichowicz 1983). Calving was studied as another type of glacier-generated signals (Qamar, 1988; Amundson et al., 2008). Later on, a temporal seismic and acoustic calving monitoring with use of receivers located in close proximity of calving fronts was introduced (O'Neel et al., 2007; Richardson et al., 2010; Głowacki et al., 2015). Those studies revealed a diversity of calving-related signals characteristics depending on the type of calving event. Authors agree that it is possible to determine calving rate on the basis of the recorded seismic events and to obtain information about waveform parameters of those events. Finally, ice-vibrations were identified as another type of glaciers' seismic activity, likely caused by large-scale ice dynamic processes (Górski and Teisseyre, 1991; Górski, 2004). The characteristics of the signals generated by ice-vibrations vary between different glaciers mainly because of the glaciers' size.

The amount of seismic events generated by glaciers is changing greatly during the year, having its peak during late summer (Ekström et al., 2006; Bartholomaus et al., 2015). Those seasonal changes can be studied by the dedicated local seismic networks targeting detection and location of the glacier-induced seismicity (e.g. Koubova, 2015), however, due to their temporary deployment, those networks cannot capture long-term changes. On the other hand, there is a wealth of the seismological data available from the stations located in the polar regions, recording continuous data streams over years. Some of those stations are located in the proximity of glaciers, hence making the detection of the glacier-induced seismic events feasible (Köhler, 2015). However, in order to use this data in the context of studying glacial dynamics, new processing workflows need to be developed and implemented, because standard seismological approaches are set up for another type of events with different characteristics (teleseismic or regional earthquakes).



In this paper we develop an algorithm for automatic seismic event detection and classification
based on parameters that are easy to derive from the continuous seismological records. Our
results show long-term glacier-related seismic activity in the vicinity of both Hansbreen and
Kronebreen glaciers in Spitsbergen. We present annual variations, as well as significant
increase in number of detected events between 2008 to 2014.
First, we introduce the datasets used in this paper. Then, we describe our detection procedure
and classification algorithm based on fuzzy logic. Finally, we analyse monthly and year-to-
year glacier-induced seismicity distribution together with the meteorological data for both
stations.

## 2  Data

We use continuous seismic records from the seismological broadband station called HSPB,
located in the Hornsund fjord near the Polish Polar Station in southern Spitsbergen, as our
primary dataset. The HSPB station belongs to the Polish Seismological Network (PL)
operated by the Institute of Geophysics, Polish Academy of Sciences. In 2007, within the
International Polar Year activities, the station was upgraded with the help of the NORSAR
organization and the broadband (STS-2) seismometer was installed. It enabled analysis of
distant (teleseismic) earthquakes (Wilde-Piórko et al. 2009).
The HSPB station is located in the proximity of the Hansbreen glacier (Fig. 1). The distance
to the terminus is about 3 km. Hansbreen is a polythermal tidewater glacier. It has a surface of
56 km$^2$ and is 16 km long. Its frontal cliff is about 1.5 km wide and 30 m high above the sea
level (Błaszczyk et al., 2009, Grabiec et al., 2012).
The second dataset comes from the broadband station called KBS (IU network), located in the
Kings Bay, Ny-Ålesund, western Spitsbergen (Fig. 1). The station is operated by NORSAR
and also equipped with the same broadband seismometer (STS-2).
There are a few major glaciers in the area (Kronebreen, Kongsvegen, Kongsbreen) and a
number of minor ones. The distance to the combined terminus of Kronebreen and
Kongsvegen is about 13 km. The Kronebreen's surface is 445 km$^2$ (Trusel et al., 2010) and it
significantly contributes to the overall area of local glacier system.
Both datasets are freely available online, e.g. through the IRIS DMC
(http://ds.iris.edu/ds/nodes/dmc/). In case of the HSPB station, we used the data recorded
between 2008 and 2014, available in the online Orfeus database (http://www.orfeus-eu.org/).



Due to the lack of data in last quarter of 2009 caused by maintenance works, we used the last
quarter of 2007 instead.
In case of the KBS station we processed the data available between 2010 and 2014. The
analysed period differs from the HSPB station, due to the lack of the continuous raw data in
the online IRIS database.
## 3   Data processing
Our data treatment scheme is aimed at automation of processing of large continuous data
volumes in the context of detecting and classifying glacier-induced seismic events. The
sequence of processing procedures and its parameters were chosen in the way to produce an
autonomous processing sequence, easy to implement for any dataset. Each dataset was
processed with strictly the same procedures and parameters in order to provide results which
would not be biased by data processing.
### 3.1   Basic event detection workflow
We start our data processing workflow with event detection algorithm. All the data was
bandpass-filtered with cut-off frequencies 1 and 15 Hz. Such cut-off frequencies were chosen
to remove low-frequency microseisms, generated by ocean waves, and high-frequency noise.
For the filtered data, in a moving window, we calculated the ratio of the short term average to
the long term average of the signal (STA/LTA). Then, the events' duration times were
calculated for all three components of the recorded data (i.e. vertical and two horizontal
components of the ground motions) based on the normalized cumulated energy density (NED)
function. The NED function was computed after subtracting a noise function (NF) from the
signal (Eq. 1).
$$NED(t) = \sum_{n=0}^{t} \left( |x(n)| - NF(n) \right) \qquad (1)$$
The noise function (NF) is a linear function, which fits cumulated noise calculated in time
window where no event occurred. In order to precisely describe the noise level, which can
vary from day to day, the noise function was calculated for each daily record separately. The
time needed for NED to arise from value of 0.15 to 0.85 was considered as event's duration
time.



Each detected event was saved as 50-second long time-series consisting of the three channels
equivalent to the recorded three-component waveform.
Subsequently, events which fulfilled any of the three elimination conditions were discarded as
obviously false detections. These conditions were as follows:
1. Duration longer than 25 s: to discard strong tectonic earthquakes
2. No variability in the frequency spectra (difference between mean and maximum

7       amplitudes for the whole detection lower than 10% of the mean value): to discard

8       detections caused by temporal noise increase.

3. Less than 10 s difference between two consecutive detections: to avoid detecting the
same event multiple times.
Such a detection procedure resulted in the total amount of 8876 detections between 2008 and
2014 for HSPB station (Fig. 2). However, except glacier-triggered events, this dataset
contained also tectonic earthquakes and false detections, triggered by, e.g. human activity
near the Polish Polar Station, which were too complex to be excluded by the above
elimination conditions. Therefore, to distinguish between non-glacier- and glacier-induced
seismic events, we developed a fuzzy logic algorithm. The algorithm was based on four input
parameters calculated for each registered event:
1. The ratio of difference between mean and maximum values of smoothed amplitude in
the frequency band 1-5 Hz to the corresponding difference in 6-10 Hz band.
2. The ratio of difference between mean and maximum values of smoothed amplitude in
the frequency band 1-5 Hz to the corresponding difference in 11-15 Hz band.
3. The number of times when the smoothed amplitude exceeds its mean value
4. The total time of smoothed amplitude exceeding its mean value for longer than 5 s.
**3.2  Fuzzy logic event classification**
The essence of fuzzy logic is to use logical variables, ranging between 0 and 1, instead of
using standard Boolean (two-valued) algebra. Hence, the fuzzy approach determines to what
degree conditions are fulfilled instead of returning yes-or-no answers. As a result one gets
membership functions which say to what degree each object, characterised by chosen
variables, belongs to each of the predefined groups with unsharp boundaries (Zadeh, 1965).





The four previously described parameters were used as input data to the fuzzy logic
algorithm, which classified all detections into four groups. Classification criteria were chosen
based on event analysis, to remove false detections and maximize a match for earthquakes and
ice-vibration groups. Events which were not clearly recognized as earthquakes, ice-vibrations
or noise were collectively marked as "not identified".
We used the following event classes with their respective characteristics:
1. Tectonic earthquake – strong and steady energy flow, which, after exceeding mean

8       value once, remains above it for at least 15 seconds.

2. False detection – strong and short energy bursts exceeding mean value more than at

10       least 7 times in a 50-second-long record.

3. Ice-vibration – signals with dominant frequency band 1 – 5 Hz, lasting from a few to

12       over a dozen of seconds.

4. Not identified – signals not matching any of the characteristics described above.
The fuzzy logic algorithm starts with the evaluation how input parameters, individually for
each of the events, satisfy the criteria by which event classes are characterized. Then it
chooses the event class which is suited best by the parameters. The graphical representation of
this idea with an exemplary event classification is presented in Fig. 3.
The outcome of fuzzy logic distribution for the HSPB data is presented in Fig. 4. The majority
of events were categorised as "not identified". However, the temporal distribution of the
events devided into groups (Fig. 5) shows, that the "not identified" group tends to follow
strictly the year-long glacier seismic activity pattern (Jania et al., 1985; Ekström et al., 2006;
Bartholomaus et al., 2015). It suggests that most of these events are actually glacier-induced,
but their waveforms differ from waveforms of low-frequency ice-vibrations. Further in this
study we treat them as glacier-induced. Hence, the remaining two groups of events, "tectonic"
and "false", are not glacier-induced and are excluded from further anylysis.
In case of the data from the HSPB station, the fuzzy logic classification algorithm resulted in
7020 detections considered as glacier-induced and 1858 detections in the tectonic or false
groups.
In order to further test our classification algorithm, we applied the same workflow to the data
from the KBS station recorded between 2010 and 2014, acquiring 17711 detections. Then the



same fuzzy logic classification procedure was carried on, resulting in 2798 events classified
as tectonic or false and 14913 events classified as glacier-induced.

## 3   4   Results

### 4   4.1   HSPB dataset

Our results clearly illustrate changes in the long-term Hansbreen glacier seismic activity.
Figure 6a shows the periodicity of glacier-induced events occurrence and year-to-year
relation. The monthly distribution follows the same pattern each year. During winter and
spring it stays at base-level activity, then, it intensifies from June to November, having its
peak in August and September. We have found this scheme true for all analysed years except
2011. That year, the typical events distribution was slightly blurred and the amount of events
in July and August, as compared to June and September, significantly decreased.
The monthly detection distribution summed over the analysed period (2008-14) is shown in
Fig. 6b along the mean temperature and summed precipitation curves. We observe a one
month delay between the temperature and events' count peaks. This delay can be interpreted
as the time needed by such an enormous ice mass to start a reaction to the temperature
growth. The correlation coefficient (Pearson's) between monthly event distribution and mean
month temperature equals 0.79, while the one calculated with a one-month lag equals 0.85. If
we consider Positive Degree Day (PDD) measure, we obtain an even higher correlation
coefficient (0.95) with similar one-month lag. In case of the summed precipitation data, we
obtained the correlation coefficient of 0.82 (with no time lag observed).
Figure 6c presents the total number of glacier-induced events every year since 2008. It shows
mean temperature and summed precipitation each year, but only in the period between June
and November, which is the most important period in terms of glacier activity. We notice
almost a double increase in the amount of events in 2013 as compared to previous years. It is
accompanied by the noticeably steady growth of the mean temperature in warm months
(June-November) of 1.5°C in the analysed 7 years period. Although the correlation
coefficients for monthly distribution were very high, they severely decreased for year-to-year
data. The coefficient for the mean temperature decreased to 0.56, while coefficients for
summed precipitation and PDD index decreased nearly to 0.




## 4.2   KBS dataset

In case of the KBS dataset, despite the two-year shorter time span, we detected over two times more glacier-induced tremors as compared with the HSPB dataset. Figure 7a presents distribution of those events. We observe a steady increase of number of the events from year to year.

Similarly to HSPB dataset, monthly distribution of glacier events near Ny-Ålesund correlates well with Positive Degree Day index and summed precipitation (Fig. 7b), reaching the correlation coefficient of 0.96 (with one month lag) and 0.91 (no lag), respectively. The correlation coefficients for year-to-year event distribution with mean temperature and precipitation in warm months also decrease to 0.61 for mean temperature and to 0.39 for precipitation (Fig. 7c). However, contrary to the Hornsund dataset, we observe higher correlation coefficients for whole year mean temperature (0.88) and for whole year summed precipitation (0.91).

## 5   Discussion

There is a significant disproportion between the two presented multiyear distributions of the glacial seismicity in the two different regions of Spitsbergen. Results from the Kings Bay region affiliated with large Kronebreen glacier system, regardless of the shorter time span, show nearly doubled detections number as compared with the Hornsund dataset. Reasons for such disproportion can be explained in terms of many factors contributing to the total detection number. First of all, there is a much larger surface of glaciers surrounding the KBS station. Also, the background noise level, which determines the smallest events possible to detect, is different between the two locations. The STA/LTA trigger was adjusted for the more noisy HSPB dataset, implying that small events from KBS are more likely to be detected because of their bigger signal-to-noise ratio. Furthermore, the dynamics of the glaciers themselves can differ, e.g. the width of the Kronebreen terminus is two times wider than Hansbreen's, implying a higher calving rate. In addition, 5 km from its terminus, the Kronebreen interacts with the Kongsvegen glacier (Trusel et al., 2010) generating seismic events at the interaction zone (Koubova, 2015).

The observed delay of the peak seismic activity with respect to the temperature and the PDD supports Lackman et al. (2015) inferences that it is not the air, but the sea temperature at the terminus that has the key impact on the calving rate. Consequently, the difference in the overall seismic activity between the analysed locations partially caused by different calving





rates can be linked to the differences in the glacier geographical exposure to sea circulation.
The ocean temperature on the western Spitsbergen coast can vary significantly, depending on
the actual range of the West Spitsbergen Current (Walczowski et al., 2009).
The potential weakness of our method is the lack of the events' epicentral (i.e. spatial)
locations, which are hardly possible to obtain using a single station. Without this knowledge
we cannot affiliate tremors with any particular glacier. Consequently, we affiliate them with
the station area, i.e. with the largest ice masses surrounding the station.
The computed duration time of each event is shorter than its true duration time, which is the
consequence of the method we used. However, this parameter is used mostly as a reference
parameter to compare various events, not to determine the factual event duration. Events of
the durations exceeding 25 s were excluded from the beginning of the data processing
procedure, which was intended to eliminate noisy detections. Very long and very low-
frequency ($< 1$ Hz) tremors were not analysed in this study.
The fuzzy logic system was designed using characteristic events recorded by the HSPB
station. Its main task was to separate tectonic and noisy signals from the glacier-induced ones,
provided that the false detections (e.g. caused by human activity), tectonic earthquakes and
the glacier origin tremors are the only sources of seismic signals.
The robustness of the algorithm is satisfactory. It needs about one day to process a few-year-
long continuous data on a PC computer. However, the parametrization of the grouping
conditions for the fuzzy logic algorithm can differ for various datasets because of different
factors such as: noise level, distance to the sources in a glacier, human activity, and others. It
requires a preliminary study on a sample data set to adjust the parameters for a given location.
The smallest variability of parameters is expected in the "earthquakes" category, because
earthquakes have similar signal characteristics everywhere.
The standalone grouping algorithm might also be used in multi-station glacio-seismological
research to produce preliminary event origin classification, which can significantly decrease
the number of non-glacier-induced signals in further analysis.
**6    Conclusions**
In this study we used continuous seismological data recorded by the two broadband stations:
HSPB in southern Spitsbergen and KBS in western Spitsbergen to analyse glacial seismic
activity near the Hansbreen and Kronebreen glaciers. We designed a special detection



workflow together with an event classification algorithm. The grouping algorithm operates
using fuzzy logic and distributes detections among four groups: ice-vibrations, tectonic
earthquakes, noise and not identified events.
We detected and classified over seven thousand events throughout a 7-year-long time span
(2008-2014) in the HSPB station region and over seventeen thousands throughout 5-year-long
time span (2010-2014) for the KBS dataset.
The main conclusion of this study is that over recent years the glacier-related seismicity in the
analysed regions of Spitsbergen increased significantly. For last two years (2013-14) the
number of the glacier-origin events for the HSPB dataset was doubled. For the KBS dataset
we have observed a steady increase of number of events from year to year.
The monthly events distribution summed over analysed period correlates well with the
seasonal temperature variations. The highest correlation coefficients (0.95 and 0.96) were
observed between the glacier seismic activity and the Positive Degree Day (PDD) index
delayed by one month for both datasets. Correlation coefficients with the mean temperature
and summed precipitation were also very high. A year-to-year distributions reveal much
weaker correlations or no correlations.
Our results indicate the promise of using long-term seismological observations from the
permanent polar seismic stations located in proximity to glaciers to study their associated
seismic activity. With the help of the event detection and grouping algorithm the number of
the glacier-generated tremors can be assessed, showing temporal changes and long-term
trends in glaciers' dynamics.
**7    Acknowledgements**
The HSPB seismological station is operated by the Institute of Geophysics, Polish Academy
of Sciences, in cooperation with NORSAR research foundation. It was installed within the
framework of an IPY project, mainly financed by the Research Council of Norway (contract
no. 176069/S30), and is part of Polish Seismological Network. The KBS seismological station
belongs to the Norwegian Seismological Network and is maintained by the University of
Bergen. We would like to thank Tomasz Wawrzyniak and the Hornsund Polish Polar Station
staff for establishing and maintaining the Hornsund GLACIO-TOPOCLIM database used in
this investigation. We would also like to acknowledge the Norwegian Meteorological Institute
for establishing and maintaining the free access weather and climate data eKlima service used





in this paper. Seismological records were downloaded from free online databases of IRIS and
Orfeus. This work was partially supported within statutory activities No 3841/E-41/S/2015 of
the Ministry of Science and Higher Education of Poland.

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



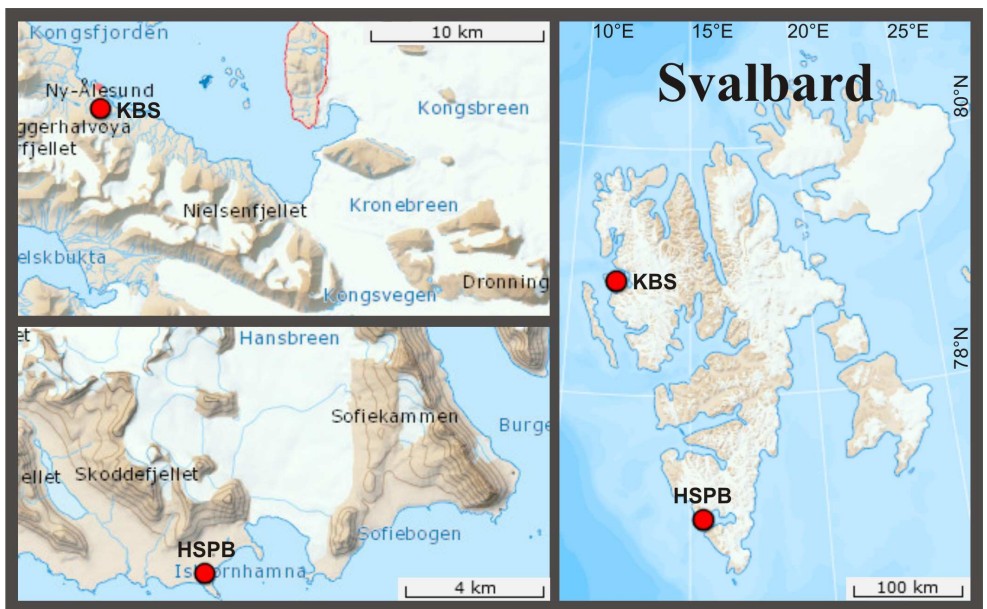

*Figure 1. Location of seismological stations and nearby glaciers a) HSPB station and Hansbreen glacier in the Hornsund fjord; b) KBS station and Kronebreen glacier in Ny-Ålesund. Modified from online Map of Svalbard: http://toposvalbard.npolar.no/.*

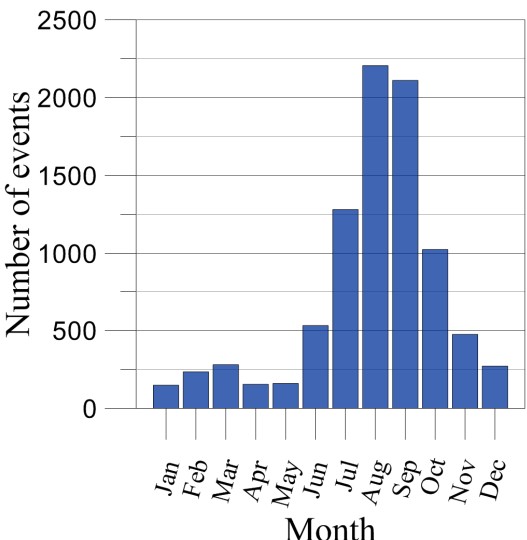

*Figure 2. Monthly distribution of all detections from the HSPB station in years 2008-2014.*



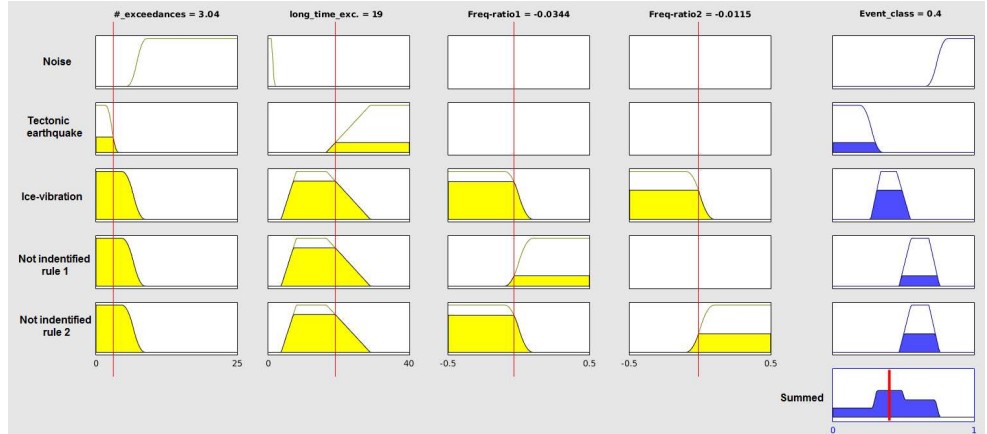

*Figure 3. The graphical representation of fuzzy logic rules evaluation in the classification algorithm. The rules characterizing event classes are displayed in rows, the input parameters in columns. Exemplary input parameters' values – thin red solid line. Yellow fields indicates to what degree each partial condition is fulfilled by values of the exemplary input parameters. Blue fields indicate to what degree input parameters fulfil rules characterizing each event class. The rule fulfilled the best constitutes the result – in this case the event was classified as ice-vibration – thick red solid line.*

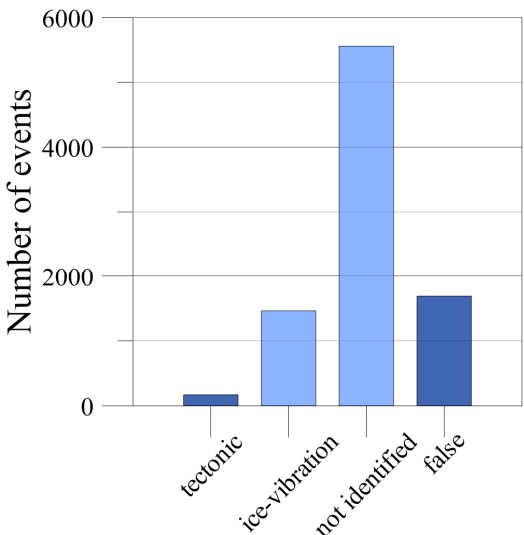

*Figure 4. The classification of events recorded in years 2008-2014 on the HSPB station. Light blue coloured bars are affiliated with the glacier-origin events.*



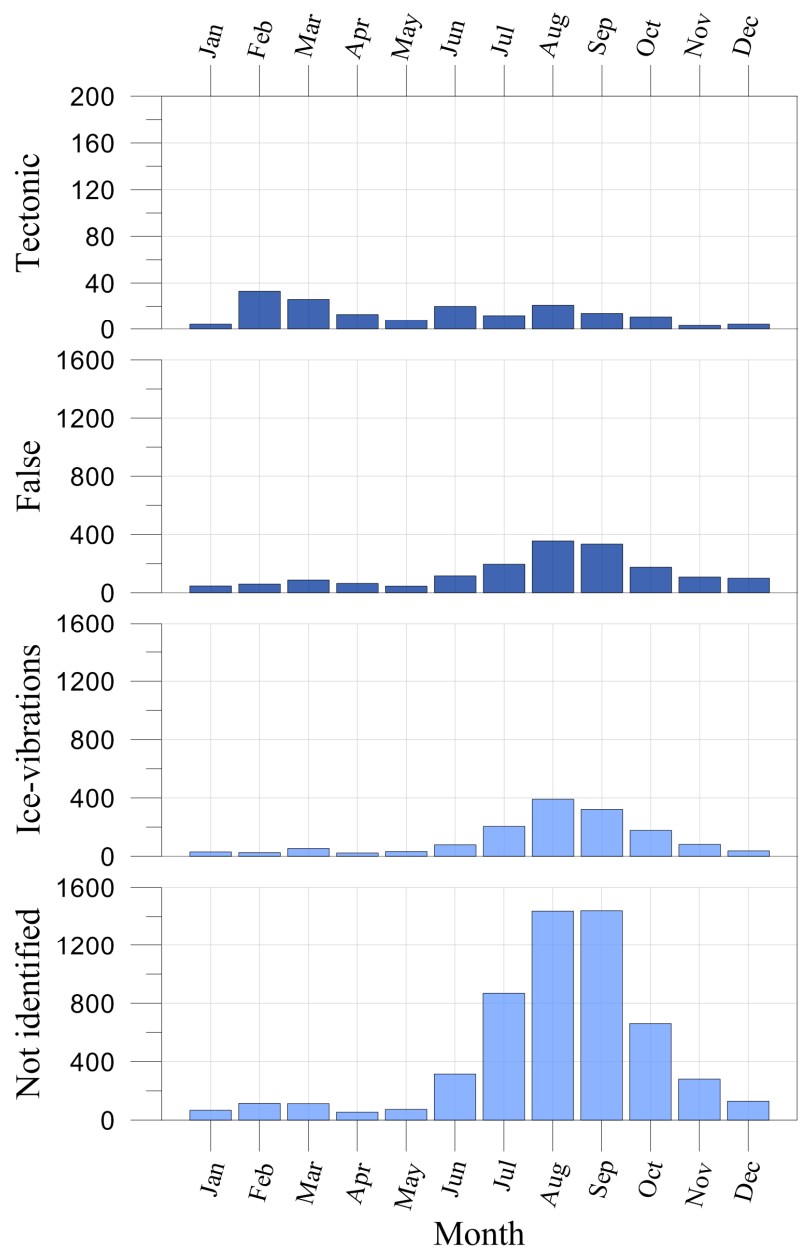

*Figure 5. Monthly distribution of events in each group from the HSPB station. Light blue coloured groups are affiliated with the glacier-origin events.*





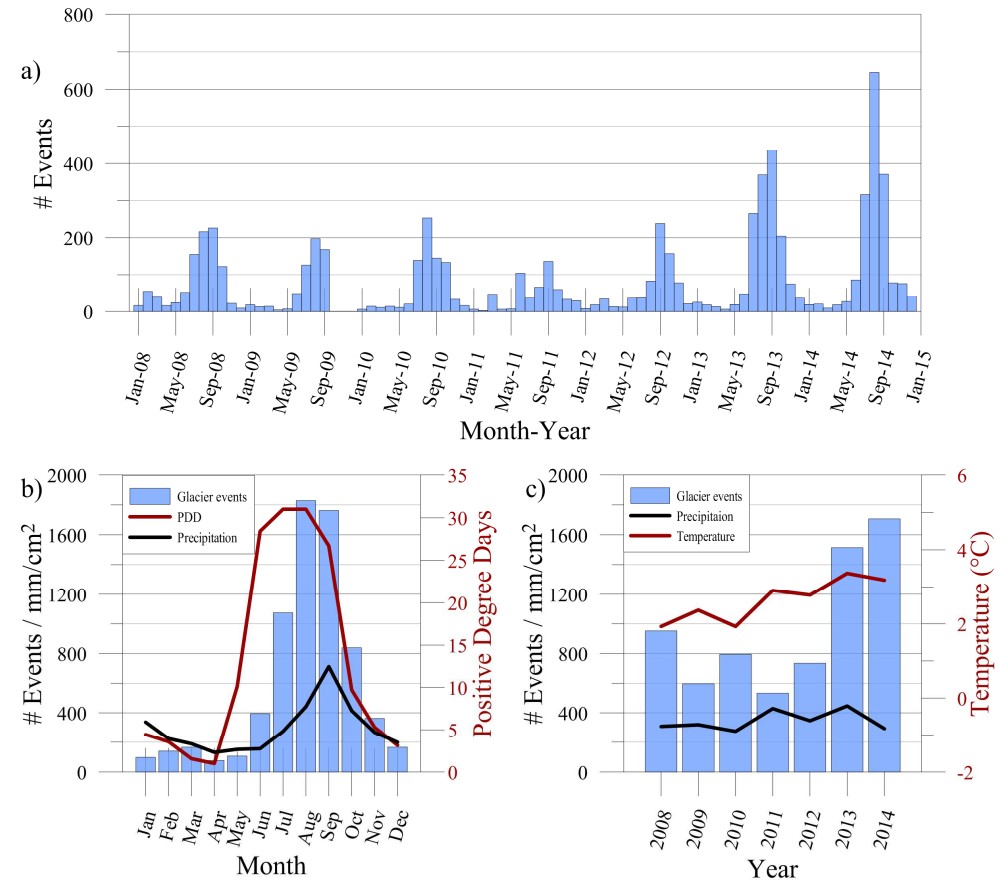

*Figure 6. Temporal distribution of glacier-induced events from the HSPB station. a) one-month step distribution; b) monthly distribution of all events summed over 2008-2014, summed precipitation – black solid line, PDD – red solid line; c) distribution of all events between 2008-2014, summed precipitation in warm months (VI-XI) – black solid line, mean temperature in warm months( VI-XI) – red solid line*





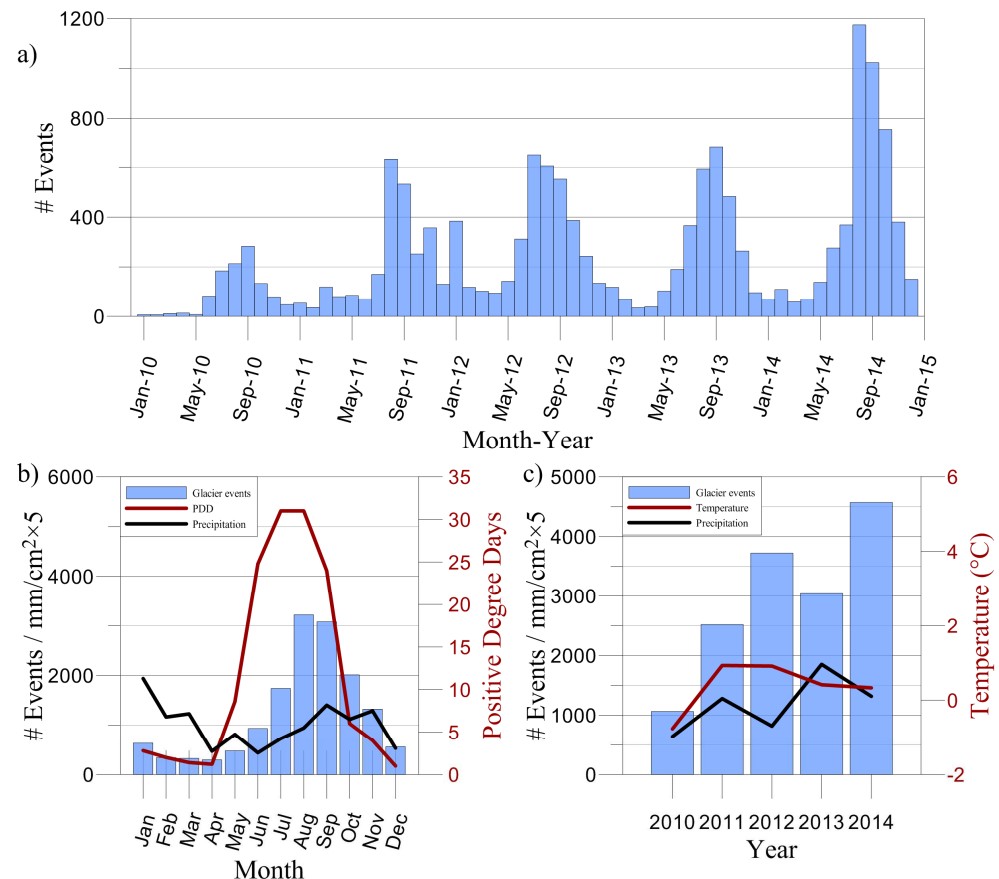

*Figure 7. Temporal distributions of the glacier-induced events from the KBS station. a) one-month step distribution b) monthly distribution of all events summed over 2010-2014, summed precipitation – black solid line, PDD – red solid line; c) distribution of all events between 2010-2014 summed precipitation in warm months (VI-XI) – black solid line, mean temperature in warm months( VI-XI) – red solid line.*