# Peer review of "Monitoring long-term changes of glacial seismic activity"

_The Cryosphere, 2015_

## Referee Comment (RC1) · Anonymous Referee #1 · 16 Feb 2016

Review of Gajek et al. "Monitoring of long-term changes of glacial seismic activity with continuous seismological observations: a case study from Spitsbergen" Under consideration for publication in The Cryosphere, 16 February 2016

This manuscript describes work classifying seismic events recorded near glaciers in Spitsbergen and describes variations in the occurrence rate of these events. The authors identify correlations between seasonal weather data and the seasonal occurrence of these events and an increase in the number of events. Their method to identify the origin of detected seismic events is new to glacier seismology.

Despite these efforts, there are a variety of issues with the present manuscript that I suggest the authors resolve prior to publication. These items limit the reproducibility

of the work, the originality and significance of the conclusions, and the extent to which the work can be understood. My major concerns, which I expect will require significant time and effort, are briefly summarized below, with more minor line edits following. Following significant revision, a new manuscript may be appropriate for publication.

Major comments: + The methods are extremely difficult to follow. It is unclear how the NED will evolve over time, or how the noise function was calculated. The event classification criteria (in two numbered lists on p. 5) are ad-hoc and are presented without justification. The explanation of the fuzzy logic algorithm is very hard to follow and there is not nearly enough information provided to allow for interpretation of Figure 3. Inclusion of sample waveforms, illustrating the different criteria, would be of great help. My understanding is that "ice vibrations" are calving icequakes. Is this not the case?

+ There is no description of the origin of the weather data or how the positive degree days are calculated. These should be part of the methods.

+ The value of the fuzzy logic criteria is not clear, since about 60% of seismic events are not classified, nor is it clear how the different types of events differ from each other. Why do the authors believe this approach was useful? Perhaps is the method were more clear, its impact would likewise be more easily appreciated. How do the authors know to attribute the "Not identified" events to the nearby glaciers?

+ The conclusions are not new. Kohler and others (Polar Research, 2015, 34, 26178) published a paper last year drawing on the same seismic signals, using more seismometers and applying more compelling analyses to these data. Kohler and others convincingly link the icequakes to calving events and reveal a seasonal cycle nearly identical to that reported in the present manuscript. Luckman and others (Nature Communications, 2015) also produce time series of frontal ablation rates that will contain calving events with similar calving events. The present authors cite both of these studies, but it is not clear how the present work is different than or similar to these existing

studies. The authors have the opportunity to advance our understanding of calving seismicity and calving through more careful comparison to these existing studies. As it stands now, the conclusions are both weaker and more inconclusive than the conclusions of previous studies.

+ The descriptions are unnecessarily qualitative in a number of locations within the text, for example when adjectives such as "major" or "minor" are applied without definition.

+ The quality of the writing needs improvement prior to publication.

Line edits follow: p. 1 L 12: remove the first "the"

p. 1 L 13: "over many years" is redundant

p. 1 L 20: What is "energy flow analysis?" Energy of what? This is not described in the main text.

p. 2 L 16-19: Please provide more context about these "ice vibrations," since they appear throughout the present manuscript. Comparison of the Gorski literature with other papers published on glacier seismicity (by O'Neel, Bartholomaus, and Kohler) suggests that the ice vibrations might be calving icequakes.

p. 3 L 25-26: Please define what you mean by "major" and "minor" here.

p. 4 L 1-2: What do the authors mean by this?

p. 4 L 4-5: This conflicts with the earlier statement that the seismic data is available in the IRIS DMC databases.

p. 4 L 20-23: How is this an energy density? Do the authors use velocity seismograms? Subtracting the noise from the absolute value of the ground velocity doesn't make an energy.

p. 4 L 24-25: Please provide more information regarding how the noise function was calculated. How was the noise fit? What's the size of the moving window? How do you

know that no event occurred (i.e., based on what criteria)?

p. 4 L 27: It appears to me that the NED as defined in Equation 1 would increase consistently through time. I don't see how these thresholds work to trigger detections in the monotonically rising NED values. How were these thresholds chosen?

p. 5 L 5-8: What are the justifications for these criteria? Glacier-produced calving icequakes can sometimes exceed 25 s (Bartholomaus and others, 2012 and 2015, in JGR)

p. 5 L 6: What kind of variability is intended here? in the spectra, or over time?

p. 5 L 18-21: It is hard to understand what the authors intend by these sentences. How are the amplitudes smoothed?

p. 5 L 22-23: This description could be aided by an illustration.

p. 6 L 3: What kind of event analysis? How were the events analyzed?

p. 6 L 7: What is "strong and steady energy flow"? This is language not traditionally used in seismology.

p. 6 L 21: What is the "strictly year-long pattern"? Do the authors mean "seasonal"?

p. 6 L 23-25: The assumption that the "not identified" events are glacier-generated because their occurrence varies seasonally is very weak evidence. How can the reader know that they're not rockfall, or river produced, or artifacts in the data? How is "false" different than "not identified"?

p. 7 L 10: What do the authors mean by "slightly blurred?"

p. 7 L 13: Fig. 6b shows PDD, not temps. But the PDD that's shown doesn't look like other typical PDD values. The positive degree days values are the cumulative daily temperatures above 0 degrees (as described in Hock 2005 and other papers). This looks to me like the number of days per month that exceed 0 degrees.

Interactive
comment

p. 7 L 14-16: What mechanism is implicated here? This is extremely loose and imprecise language.

p. 7 L 17: Monthly temperatures are not shown. Please plot if discussed.

p. 7 L 24: "doubling" instead of "double increase"

p. 7 L 29: plot the annual PDD here.

p. 8 L 20: What are the authors implying here? What is the connection between the glaciated surface area and the number of seismic events? I believe that Kronebreen is a much faster-flowing glacier than Hansbreen. That might explain more calving at Kronebreen than at Hansbreen. What about the detectability of these signals? Are the seismic stations equidistant from glaciers? Perhaps attenuation might change the different detectability of the seismic signals.

p. 8 L 25: The glacier dynamics "do" differ, not just "can" differ.

p. 8 L 27-28: What is meant here? How do these glaciers "interact"? How do these interactions generate seismic signals? What is the proposed mechanism?

p. 8 L 30: "Luckman" instead of "Lackman"

p. 9 L 1-3: Please provide more context here with the Luckman and Kohler results. Are the authors implying that ocean temperatures might be promoting calving during the fall? What other evidence can be provided to strengthen this case? Are the results here different than the Luckman and Kohler results in some way?

p. 9 L 6: "Tremor" in seismology is a very specific type of seismic signal, see literature on volcanic tremor or tectonic tremor (and slow slip earthquakes). The authors should use a different word, such as "seismic signals."

p. 9 L 8: What is the "true" duration time? "True" according to what analysis?

p. 9 L 15: What is meant by "noisy" signals? "Noisy" in what way? It doesn't appear to

me that the fuzzy logic method provided much value.

p. 9 L 18-19: I recommend removing this sentence, but if the authors choose to retain it, please provide more information about the benchmarking experiments. What kind of computer was used to run this approach?

p. 11 L 28: typo in "micro"

Figure 3: As presented, this figure is unsuccessful in adding value to the manuscript. What is an "exemplary input parameter value?" What are the x and y axes in each panel? I don't understand what is being shown here.

Figure 5: The basis for affiliating the "not identified" events with the glacier needs more support in the text.

Figures 6: panel a: Is there an outage in the fall of 2009? This should be indicated if so. The units in black on panels b/c are unclear. It looks as though there is a complicated division taking place. Are the "mm/cmˆ2" one unit? Units of precipitation should be mm or m. The "per area" is meaningless. Roman numeral months in the caption should be replaced by the month names.

Figure 7: same problems as Figure 6.

[Figure]

---

## Referee Comment (RC2) · Anonymous Referee #2 · 18 Feb 2016

The work is interesting and it points out some important results about the seismicity in Spitsbergen (Svalbard, Norway) and its correlation with glaciers, seasons of the year and weather data. However, in my opinion it is not ready for publication since some important parts of a paper that aim to reach the broad community of Cryosphere readers are missing. In particular: a) a comprehensive description of the problem, b) a robust validation of the claimed results and discrimination between different type of events, c) a comparison with already published and similar results. For these reasons I would suggest a major revision of the manuscript. I do not enter in the details discussion and conclusion since I expect that the suggested further analysis would change these two sections.
[Figure]

Major points:

1) Introduction. Readers not familiar with Spitsbergen location and characteristics get lost from the beginning of the manuscript. It is not mentioned that this is a Island belonging to the Svalbard Archipelago (Norway). Maps in Figure 1 are never referenced in the manuscript. In the Introduction a description about Spitsenbergen is missing. I would expect a section describing why this work is focused on this region, why we expect seismic activity here, what is the size of these events and some description about previous studies about the region. Since one of the goal is also to discriminate between tectonic and "glacier related" events, I would also expect a brief description about the seismicity of the region and about the differences between the two type of events. I would expect a comparison with other detection algorithms as standard seismic detection/pickers and more specific algorithms used for glacier related events (e.g. that by Walter Olivieri Clinton, J. of Glaciology 2013)

2) Data and Analysis. The authors go straight to the technical description of the methodology but again, in my opinion, a crucial part is missing that would help the reader to understand the problem and how it has been tackled by the authors. There is not a definition of "event" and possibly some figure with seismograms and spectra for the different type of events would help the comprehension. For the case of the spectra, a reference to background noise is mandatory to identify the signal and to understand filters and thresholds used. A figure describing NED(t) and NF(t) would also help as well as a formula for NF(t).

3) Numbers. The authors describe their method without mentioning how they selected the "numbers" as for the case of the bandpass filter between 1 and 15Hz, 0.15-0.85 fr the duration, 25 seconds, "more than 7 times in 50 seconds" and so on. It is not clear if this is an a-priori choice or if it follows tests (e.g. trial and error) or data analysis on the different nature of the different signals. This would increase the reproducibility of this study and its eventual application to other regions and data, similar or slightly different.
4) Fuzzy logic classification. Description is qualitative, replication of the study is almost impossible and some crucial specification are missing. At line 7-8 of page 6 I read "strong and steady energy flow, which, after exceeding mean value once remains above it for at least 15 seconds". I guess the authors used a formula to convert the seismogram (velocity) into energy. This should be specified together with the rules to compute the "mean". If it is NED(t) it should be mentioned. In line 11 of page 6, the description of the selection rule for "Ice vibrations-signals" is also vague.

5) Tectonic earthquakes. For the case of HSPB the authors detect 1858 earthquakes and even more for KBS (2798). Does any of it appear in published catalogue? Why or why not? What is their size in terms of magnitude and why they occur? Is there any event listed in catalogues that was missed by the described detection algorithm. How the seismic sequence that interested the Storefjorden impact on the detection/discrimination process? (The Storfjorden, Svalbard, 2008–2012 aftershock sequence: Seismotectonics in a polar environment by Myrto Pirli, Johannes Schweitzer , Berit Paulsen Tectonophysics doi:10.1016/j.tecto.2013.05.010)

6) Validation of the results. The authors claim the success of the described methodology. But they do not mention if any test was implemented to validate or to cross-check their results. Common procedure, in seismology, is to visually inspect data to compare automated detection with visual observation. It is mandatory, in my opinion, a validation test, to explore the "efficiency" of the proposed methodology in terms of missed events (of the three kinds) and of false detections. I would be really surprise to see that both numbers (missed and false) are equal to zero. Paper by Kohler et al., mentioned in the Introduction, produces a similar catalogue but this is not discussed in the manuscript. Results are not compared even though I read in Kohler et al. "Most events occurred between July and December, with peak activity in August and September. Seasonal seismicity varies in accordance with expected glacier dynamic activity, ...."

7) Seasonality. A description of the variability of the background noise over months (and years) is missing. If noise changes, detection capability of small events changes

as well. This test is mandatory prior to explore the seasonality of the number of events. In seismology, the study of the rates of seismicity over time commonly relies on two concepts: magnitude completeness and declustering . The first prevents the risk of comparing the number of events in two epochs in which the detection threshold was different. The second prevents to include "aftershocks" in the analysis of time-varying rates of events. I wonder if the authors considered these (noise amplitude, completeness, clustering) to prevent a misinterpretation of the variability for the number of events over time.

8) Correlation. The authors claim that glacier-related events originated at Hansbreen and Kronebreen (Page 3, line 4) and later they mention they could not locate them (Page 9, line 4-7). Single station location techniques exist even though they are difficult to implement and these should at least mentioned and discussed. I would remark that a paper was published on this topic (http://link.springer.com/article/10.3103/S0747923915030032) using data from HSPB. This paper is surprisingly not cited. Moreover, I remain convinced that a proof about the relation between the observed events and the glaciers' activity is missing. For the case of Greenland, for example, such correlation has been found on the base of further observations as filming or water pressure data. Any further source of "earthquake like" signal is present in the region? Plants, Mines, Dams and so on?

Further issues:

- type of used filters is not described (Butterworth?).

- it is the combination of seismometer+digitizer that gives a broadband response.

- the last quarter of 2007 was included in the analysis but the time-span for the results from HSPB dataset is always referred as (2008-2014). How this affects figure 6a and 6c?

- As far as I could see, HSPB data at Orfeus data-center start in Jan 2010, am I wrong?

[Figure]

- Page 5, line 5, this selection rule aim to discard "strong tectonic earthquakes", I wonder if the authors refer to those occurring at regional and/or teleseismic distance. This should be specified because in the following of the manuscript they count the detected tectonic earthquakes.

- Page 6, line 27, 7020+1858 = 8878 while at page 5 line 11 the number of detected events is 8876

- Figures are sometimes references as "Fig." and sometimes as "Figure"

- Page 7, line 5. I would suggest to first describe the result and then comment on them.

- As far as I could see on EIDA server at Orfeus, two further stations exist in the region. Could their data help the discrimination and location of part of the events?

- Figure 1, I would suggest to reference the two maps on the left on the right map, to help the reader. Furthermore, I would suggest to write only the relevant toponyms to ease their identification on the map.
* * *

---

## Author Comment (AC1) · 17 Apr 2016

*Dear Reviewer,*

*We would like to thank you for your time and efforts devoted to revise our manuscript and for all the suggestions, which we find essential for right understanding of our work. The revised manuscript, remarkably different from the original version, will be submitted to the editorial office after posting reply to your comments. Changes in the manuscript involved adding extra figures, putting more emphasis on the aims and conclusions and expanding descriptions and definitions throughout whole paper. Below we refer to all your remarks in the sequential manner.*

Anonymous Referee #1

This manuscript describes work classifying seismic events recorded near glaciers in Spitsbergen and describes variations in the occurrence rate of these events. The authors identify correlations between seasonal weather data and the seasonal occurrence of these events and an increase in the number of events. Their method to identify the origin of detected seismic events is new to glacier seismology.

Despite these efforts, there are a variety of issues with the present manuscript that I suggest the authors resolve prior to publication. These items limit the reproducibility of the work, the originality and significance of the conclusions, and the extent to which the work can be understood. My major concerns, which I expect will require significant time and effort, are briefly summarized below, with more minor line edits following. Following significant revision, a new manuscript may be appropriate for publication.

Major comments:

+ The methods are extremely difficult to follow. It is unclear how the NED will evolve over time, or how the noise function was calculated.

*Authors: We provided an illustration and broaden descriptions in adequate paragraphs, providing a reference to the literature, for a better insight into deriving the NED and noise functions.*

The event classification criteria (in two numbered lists on p. 5) are ad-hoc and are presented without justification. The explanation of the fuzzy logic algorithm is very hard to follow and there is not nearly enough information provided to allow for interpretation of Figure 3. Inclusion of sample waveforms, illustrating the different criteria, would be of great help.

Authors: *The description of the methods was significantly changed. Also sample waveforms were provided.*

My understanding is that ice vibrations are calving icequakes. Is this not the case?

*Authors: Although signal characteristic of those is similar we can't say these are the same. Ice-vibrations' sources has been localised inside the body of the glacier, close to, but not at the calving front. Also, ice-vibrations has been observed at alpine glaciers (Górski, 2004).*

+ There is no description of the origin of the weather data or how the positive degree days are calculated. These should be part of the methods.

*Authors: Those informations have been added to the manuscript.*

+ The value of the fuzzy logic criteria is not clear, since about 60% of seismic events are not classified, nor is it clear how the different types of events differ from each other. Why do the authors believe this approach was useful? Perhaps is the method were more clear, its impact would likewise be more

easily appreciated. How do the authors know to attribute the   Not identified   events to the nearby glaciers?

*Authors: Fuzzy logic section has been expanded. This method allowed to recognize part of events as not-glacier induced and separate them, and then to choose signals having characteristics corresponding to the ice-vibration events. What is crucial all this was done fully automatically and in an objective manner (although detection criteria are still subjective and based on the expert knowledge).*

*One of the arguments to link not-identified events with glaciers is that they follow the seasonal pattern. Another one is that the signals of non-glacial origin has been described by earthquake and false-detection criteria and separated with the help of the fuzzy logic algorithm. Signals which are left are different from typical noise and earthquakes waveforms, but not similar enough to ice-vibrations to be classified so. Glacier however, is a source of signals different from ice-vibrations, but much harder to specify like e.g. icequakes or different kinds of calving.*

*Because we remove most of non-glacier-induced signals we assume the rest to correspond to glacier activity as supported by their seasonal variability.*

*What's more we analysed the same time span of the data as Kohler et al. (2015) but we used single station detections. We can assume that our STA/LTA detection algorithm should detect as a minimum the same number of events as it was detected by Kohler et al. (2015), who used the SPITS array located at greater distance than the HSPB, using HSPB records only to verify detection results. In fact, as glacier-generated, we have classified even less events than they did. It indicates that criteria we used were more restrictive than those used by Kohler et al. Hence, we can assume, that our detections include mostly the same events as Kohler et al. (2015) have shown for Hansbreen glacier. And hence, we claim that what we show is a glacier-related seismic activity.*

*Those conclusions can be further confirmed by comparing the seasonal and interannual event distribution with work of Kohler et al. (2015).*

+ The conclusions are not new. Kohler and others (Polar Research, 2015, 34, 26178) published a paper last year drawing on the same seismic signals, using more seismometers and applying more compelling analyses to these data. Kohler and others convincingly link the icequakes to calving events and reveal a seasonal cycle nearly identical to that reported in the present manuscript. Luckman and others (Nature Communications, 2015) also produce time series of frontal ablation rates that will contain calving events with similar calving events. The present authors cite both of these studies, but it is not clear how the present work is different than or similar to these existing studies. The authors have the opportunity to advance our understanding of calving seismicity and calving through more careful comparison to these existing studies. As it stands now, the conclusions are both weaker and more inconclusive than the conclusions of previous studies.

*Authors: We included a comprehensive discussion on this topics.*

+ The descriptions are unnecessarily qualitative in a number of locations within the text, for example when adjectives such as   major   or   minor   are applied without definition.

*Authors: We inspected and specified descriptions and definitions throughout whole text.*

+ The quality of the writing needs improvement prior to publication.

*Authors: The manuscript has been revised to improve the overall writing quality.*

Line edits follow:

p. 1 L 12: remove the first   the

*Authors: Corrected.*

p. 1 L 13:  over many years  is redundant

*Authors: You're right, corrected.*

p. 1 L 20: What is  energy flow analysis?  Energy of what? This is not described in the main text.

*Authors: To be precise we meant variability of signal power in time. The terms used are now better explained in the revised version.*

p. 2 L 16-19: Please provide more context about these  ice vibrations,  since they appear throughout the present manuscript. Comparison of the Gorski literature with other papers published on glacier seismicity (by O Neel, Bartholomaus, and Kohler) suggests that the ice vibrations might be calving icequakes.

*Authors: Gorski suggests that ice vibrations are rather large scale processes in the glacier body than calving itself (Górski, 2014). He located them roughly using an array of seismometers in the distance from the calving front. This kind of signal were also observed at alpine glaciers, what excludes calving (Górski, 2004). They may potentially be one of the factors inducing calving.*

*We included additional informations about ice-vibrations into the manuscript.*

p. 3 L 25-26: Please define what you mean by  major  and  minor  here.

*Authors: By major glaciers we meant Kronebreen, Kongsvegen and Kongsbreen glaciers, the biggest ones in the close proximity of the KBS station, while by minor all other smaller glaciers in this area. We specified this terminology in the revised version and referred reader to the map.*

p. 4 L 1-2: What do the authors mean by this?

*Authors: We decided to use an extra 3 months of data from last quarter of 2007 when showing interannual comparisons in order to keep the length of all compared periods equal. We clarified this sentence in the manuscript.*

p. 4 L 4-5: This conflicts with the earlier statement that the seismic data is available in the IRIS DMC databases.

*Authors: We corrected this paragraph.*

p. 4 L 20-23: How is this an energy density? Do the authors use velocity seismograms? Subtracting the noise from the absolute value of the ground velocity doesn t make an energy.

*Authors: We called that parameter "energy density", because we used modified formula of Normalised Energy Density Function by Sarma (1971). That section was corrected in the revised version of the manuscript and additional description followed by references was added.*

p. 4 L 24-25: Please provide more information regarding how the noise function was calculated. How was the noise fit? What s the size of the moving window? How do you know that no event occurred (i.e., based on what criteria)?

*Authors: Paragraph describing the noise function was added to the manuscript.*

p. 4 L 27: It appears to me that the NED as defined in Equation 1 would increase consistently through time. I don t see how these thresholds work to trigger detections in the monotonically rising NED values. How were these thresholds chosen?

*Authors: This information was also included in an additional paragraph about the NED function.*

p. 5 L 5-8: What are the justifications for these criteria? Glacier-produced calving icequakes can sometimes exceed 25 s (Bartholomaus and others, 2012 and 2015, in JGR)

*Authors: The aim of this study is to assess long-term glacial seismicity. To produce a reliable automatic processing procedure we focused on typical events so counting e.g. glacier-induced events of extremely long duration times (>25 s), is out of the scope of this study, even though such events are proven to exist.*

p. 5 L 6: What kind of variability is intended here? in the spectra, or over time?

*Authors: It is spectra over time. Description of this criterion has been revised.*

p. 5 L 18-21: It is hard to understand what the authors intend by these sentences. How are the amplitudes smoothed?

*Authors: Amplitudes were smoothed by calculating a running average. We reformulated these sentences and added equations to make them easier to understand for the readers.*

p. 5 L 22-23: This description could be aided by an illustration.

*Authors: We reformulated these sentences and added equations, as well. We assume, that the content of paragraphs L 18-23 is described clearly enough to make illustrations unnecessary.*

p. 6 L 3: What kind of event analysis? How were the events analyzed?

*Authors: Criteria were adjusted for waveforms from HSPB dataset which were affiliated with one of the groups based mainly on the literature studies (Górski, 2004, Koubova 2015, Pirli et al., 2013). We corrected this paragraph.*

p. 6 L 7: What is  strong and steady energy flow ? This is language not traditionally used in seismology.

*Authors: We meant a long lasting exceedance of temporal signal power over its mean value. The terms used in the paper are now more precisely defined*

p. 6 L 21: What is the  strictly year-long pattern ? Do the authors mean  seasonal ?

*Authors: Yes, seasonal.*

p. 6 L 23-25: The assumption that the  not identified  events are glacier-generated because their occurrence varies seasonally is very weak evidence. How can the reader know that they re not rockfall, or river produced, or artifacts in the data? How is  false  different than  not identified ?

*Authors: Issue of identifying not-identified group of events as glacier induced was already addressed in 'Major comment #3'.*

*Criteria of the false group were chosen to eliminate signals significantly different from glacier generated signals identified as false detections. If the event fulfills those criteria, it is classified in the false group. If an event does not fulfill criteria of ice-vibrations, earthquakes and false group, then it is classified as not-identified.*

p. 7 L 10: What do the authors mean by  slightly blurred?

*Authors: We removed this confusing statement. Now we just point out, that year 2011 has lower amount of events than other years*

p. 7 L 13: Fig. 6b shows PDD, not temps. But the PDD that s shown doesn t look like other typical PDD values. The positive degree days values are the cumulative daily temperatures above 0 degrees

(as described in Hock 2005 and other papers). This looks to me like the number of days per month that exceed 0 degrees.

*Authors: Yes, Fig. 6b doesn't show temps. We show a number of days with positive daily mean temperature. We called this parameter "Positive Degree Day" incorrectly. Hence, we changed axis descriptions and corrected the manuscript text.*

p. 7 L 14-16: What mechanism is implicated here? This is extremely loose and imprecise language.

*Authors: We removed that statement. The interpretation of observed correlation time lag appears in the discussion section and is referenced in the literature.*

p. 7 L 17: Monthly temperatures are not shown. Please plot if discussed.

*Authors: The distribution of monthly mean temperatures is very similar to PDD (already changed to "the mean number of days in each month with positive mean temperature"), but has a lower correlation coefficient. Hence, to keep the figures clear and legible, we decided to not plot the less correlated parameter, as it is not further analyzed.*

p. 7 L 24:   doubling   instead of   double increase

*Authors: Corrected.*

p. 7 L 29: plot the annual PDD here.

*Authors: We added a figure illustrating the correlations.*

p. 8 L 20: What are the authors implying here? What is the connection between the glaciated surface area and the number of seismic events? I believe that Kronebreen is a much faster-flowing glacier than Hansbreen. That might explain more calving at Kronebreen than at Hansbreen. What about the detectability of these signals? Are the seismic stations equidistant from glaciers? Perhaps attenuation might change the different detectability of the seismic signals.

*Authors: We added an information about different distances to the glaciers as one of the factors contributing to the difference in total amount of detected glacier-generated events in both datasets. Being aware of all mentioned differences, we only point out the disproportion and its possible reasons and do not imply different seismic activities of glaciers.*

p. 8 L 25: The glacier dynamics   do   differ, not just   can   differ.

*Authors: Right, corrected.*

p. 8 L 27-28: What is meant here? How do these glaciers   interact ? How do these interactions generate seismic signals? What is the proposed mechanism?

*Authors: at the junction of two interacting glaciers, friction can lead to stress accumulation. Koubova (2015) proves, that at this junction some seismic events occur. How those shocks are generated and what is their mechanism is an interesting question but this is beyond the scope of this study.*

p. 8 L 30:   Luckman   instead of   Lackman

*Authors: Corrected.*

p. 9 L 1-3: Please provide more context here with the Luckman and Kohler results. Are the authors implying that ocean temperatures might be promoting calving during the fall? What other evidence can be provided to strengthen this case? Are the results here different than the Luckman and Kohler results in some way?

*Authors: Our results also show an usual 1-2 months delay between the peak event number and the peak temperature observed for both datasets. Hence, they can be treated as another proof supporting the hypothesis of ocean temperature being a dominant factor in calving mechanism with more significant impact than an air temperature. We expanded the discussion of suggested papers in the manuscript.*

p. 9 L 6:  Tremor   in seismology is a very specific type of seismic signal, see literature on volcanic tremor or tectonic tremor (and slow slip earthquakes). The authors should use a different word, such as   seismic signals.

*Authors: We agree. Corrected.*

p. 9 L 8: What is the   true   duration time?   True   according to what analysis?

*Authors: We meant true as a factual, absolute duration time of this phenomena. We reformulated the confusing sentence.*

p. 9 L 15: What is meant by   noisy   signals?   Noisy   in what way? It doesn t appear tome that the fuzzy logic method provided much value.

*Authors: That's a mistake. We changed 'noisy signals' to 'false detections'. The fuzzy logic issue was already addressed in 'Major comment #3'.*

p. 9 L 18-19: I recommend removing this sentence, but if the authors choose to retain it, please provide more information about the benchmarking experiments. What kind of computer was used to run this approach?

*Authors: We decided to reformulate it, but we keep it though. Its goal is to point out that there is no need to employ computer clusters to perform such analysis using our algorithm. It can be done in the reasonable time using a PC class computer and hence, it can be easily implemented as a routine tool for real- or near-real-time processing.*

p. 11 L 28: typo in   micro

*Authors: Corrected.*

Figure 3: As presented, this figure is unsuccessful in adding value to the manuscript. What is an   exemplary input parameter value?   What are the x and y axes in each panel? I don t understand what is being shown here.

*Authors: This figure has been rearranged and serves for a better understanding of the fuzzy logic algorithm workflow.*

Figure 5: The basis for affiliating the   not identified   events with the glacier needs more support in the text.

*Authors: The discussion on this topic was expanded. (this issue was already addressed in 'Major comment #3')*

Figures 6: panel a: Is there an outage in the fall of 2009? This should be indicated if so. The units in black on panels b/c are unclear. It looks as though there is a complicated division taking place. Are the   mm/cm^2   one unit? Units of precipitation should be mm or m. The   per area   is meaningless. Roman numeral months in the caption should be replaced by the month names.

Figure 7: same problems as Figure 6

*Authors: We edited those plots according to your suggestions.*

*References:*

*Górski M.: Predominant frequencies in the spectrum of ice-vibration events. Acta Geophysica Polonica 52 (4), 657 457–464, 2004.*

*Górski M. 2014 – Seismic events in glaciers. Springer, 2014.*

*Sarma S. K.: Energy flux of strong earthquakes, Tectonophysics 11(3), 159-173, 1971.*

*Pirli M., Schweitzer J. and Paulsen B.: The Storfjorden, Svalbard, 2008–2012 aftershock sequence: Seismotectonics in a polar environment, Tectonophysics, 601, 192-205, 2013.*

---

## Author Comment (AC3) · 17 Apr 2016

*Dear Reviewer,*

*We really appreciate your time and efforts put in the review of our paper. We found it very helpful and believe that the suggested changes will make our manuscript much more readable and clear. The revised manuscript, remarkably different from the original version, will be submitted to the editorial office after posting reply to your comments. Changes in the manuscript involved adding extra figures, putting more emphasis on the aims and conclusions and expanding descriptions and definitions throughout whole paper. Below we list detailed answers to your comments.*

Anonymous Referee #2

The work is interesting and it points out some important results about the seismicity in Spitsbergen (Svalbard, Norway) and its correlation with glaciers, seasons of the year and weather data.

However, in my opinion it is not ready for publication since some important parts of a paper that aim to reach the broad community of Cryosphere readers are missing. In particular:

a) a comprehensive description of the problem,

b) a robust validation of the claimed results and discrimination between different type of events,

c) a comparison with already published and similar results.

For these reasons I would suggest a major revision of the manuscript. I do not enter in the details discussion and conclusion since I expect that the suggested further analysis would change these two sections.

Major points:

1) Introduction. Readers not familiar with Spitsbergen location and characteristics get lost from the beginning of the manuscript. It is not mentioned that this is a Island belonging to the Svalbard Archipelago (Norway). Maps in Figure 1 are never referenced in the manuscript.

*Authors: We added an information about Spitsbergen localisation and referenced Figure 1 in the Introduction section now.*

In the Introduction a description about Spitsenbergen is missing. I would expect a section describing why this work is focused on this region,

*Authors: Most of the studies of the glacier-related seismicity that we refer to were done in the Spitsbergen area. Thus, it was natural to choose the same region for the purpose of straightforward comparison. Also, Spitsbergen is a region that is relatively well studied with regard to tracing temporal response of the Arctic bio- and cryosphere to the changing climate. It is now stated more clearly in the manuscript.*

why we expect seismic activity here, what is the size of these events and some description about previous studies about the region. Since one of the goal is also to discriminate between tectonic and "glacier related" events, I would also expect a brief description about the seismicity of the region and about the differences between the two type of events.

*Authors: We added a description of regional and local seismicity of the Svalbard Archipelago, although tectonic earthquakes are treated by us rather as a disturbance. Comparison of glacier-generated signals, tectonic earthquakes and noise waveforms was also included.*

I would expect a comparison with other detection algorithms as standard seismic detection/pickers and more specific algorithms used for glacier related events (e.g. that by Walter Olivieri Clinton, J. of Glaciology 2013)

*Authors: In the updated version we compare our algorithm to the one by Walter et al. 2013.*

2) Data and Analysis. The authors go straight to the technical description of the methodology but again, in my opinion, a crucial part is missing that would help the reader to understand the problem and how it has been tackled by the authors. There is not a definition of "event" and possibly some figure with seismograms and spectra for the different type of events would help the comprehension.

*Authors: We've significantly changed these sections taking all above suggestions into consideration and therefore we hope that they are more clear now.*

For the case of the spectra, a reference to background noise is mandatory to identify the signal and to understand filters and thresholds used.

*Authors: The following figure (Fig. 1) is illustrating frequency spectrum of a daily record (HSPB station). We eliminate microseisms by applying a bandpass filter with lower cut-off frequency of 1 Hz, similar to e.g. Kohler et al. (2015). We chose to use higher cut-off frequency of 15 Hz based on literature study. The higher cut-off frequency is varying between different papers (Górski, 2004, Kohler et al., 2015, O'Neel Pfeffer, 2007, O'Neel et al., 2007, Walter et al., 2010) from 10 to 19 Hz. We included these informations and additional references in the corrected manuscript.*

[Figure]

**Fig. 1. Frequency spectrum of a daily record (HSPB station) at vertical component.**

A figure describing NED(t) and NF(t) would also help as well as a formula for NF(t).

*Authors: We provided an illustration and broaden descriptions in adequate paragraphs, providing a reference to the literature (Sarma, 1971).*

3) Numbers. The authors describe their method without mentioning how they selected the "numbers" as for the case of the bandpass filter between 1 and 15Hz, 0.15-0.85 fr the duration, 25 seconds, "more than 7 times in 50 seconds" and so on. It is not clear if this is an a-priori choice or if it follows tests (e.g. trial and error) or data analysis on the different nature of the different signals. This

would increase the reproducibility of this study and its eventual application to other regions and data, similar or slightly different.

*Authors: The choice of the band-pass filter limits is now backed by literature. Limits of the mNED function are also commented in the revised manuscript. We aim at detecting the glacier-induced events and then studying the long-term changes in glacier activity. Therefore to build a reliable statistics we focus on typical events, so e.g. counting glacier-induced events of extremely long duration times (>25 s) is out of the scope of this study, even though such events are proven to exist.*

*In some cases we do not explain origins of used number, because they come from strictly technical issues,, e.g. 50 seconds is a record length of each event in the database. It is twice the maximum event length we accept, to make sure it contain also pre- and post-event background noise. We believe, those have not to be necessarily included in the manuscript. However, we changed the manuscript, adding informations about some of used numbers or referencing the literature.*

4) Fuzzy logic classification. Description is qualitative, replication of the study is almost impossible and some crucial specification are missing. At line 7-8 of page 6 I read "strong and steady energy flow, which, after exceeding mean value once remains above it for at least 15 seconds". I guess the authors used a formula to convert the seismogram (velocity) into energy. This should be specified together with the rules to compute the "mean". If it is NED(t) it should be mentioned.

*Authors: Fuzzy logic description was thoroughly revised and expanded. All the rules were reformulated and supported by equations for better understanding.*

In line 11 of page 6, the description of the selection rule for "Ice vibrations-signals" is also vague.

*Authors: We formulated that statement in more clear way.*

5) Tectonic earthquakes. For the case of HSPB the authors detect 1858 earthquakes and even more for KBS (2798). Does any of it appear in published catalogue? Why or why not? What is their size in terms of magnitude and why they occur? Is there any event listed in catalogues that was missed by the described detection algorithm. How the seismic sequence that interested the Storefjorden impact on the detection/discrimination process? (The Storfjorden, Svalbard, 2008âAŽÄì2012 after-shock sequence: Seismotectonics in a polar environment by Myrto Pirli, Johannes Schweitzer , Berit Paulsen Tectonophysics doi:10.1016/j.tecto.2013.05.010)

*Authors: We found 1856 (corrected number) of tectonic or false detections for the HSPB dataset, with only 169 among them being of tectonic origin (according to fuzzy logic classification). We didn't give an exact number, however the distribution was plotted in Fig. 3. For the KBS dataset we found 351 tectonic events and 2447 false detections. Exact number is irrelevant since majority of them has been discarded during the detection procedure. Those 169 events recognized as tectonic earthquakes are only those which were not discarded by brute criterions of the detection procedure.*

*As tectonic earthquakes are beyond the aim of this study, we didn't perform magnitude nor localisation analysis. We treat them as signals which only have to be separated from glacier-origin signals.*

*Regarding the Storefjorden sequence: this figure (Fig. 2) shows distribution of events from the HSPB dataset recognized as tectonic earthquakes by fuzzy logic algorithm:*

[Figure]

**Fig. 2 Temporal distribution of events from the HSPB dataset recognized as tectonic earthquakes**

*A distinct peak can be seen during Storefjorden earthquake origin time, and its aftershock sequence can also be seen as gradually decreasing number of events. There are no significant peaks when looking at time distribution of other groups of events. However, we treat this issue being beyond the scope of this study, therefore we do not include this chart nor the Storefjorden sequence explanation to the manuscript.*

*Some of those tectonic events appear in the NORSAR regional earthquake catalogue, but majority does not. This catalogue contains events with magnitudes above 2 (sometimes above 1), while our detection algorithm was designed to reject strong tectonic events. For those reasons we believe that the tectonic earthquakes detected by our algorithm are mostly representing events too weak to be reported by NORSAR.*

*It has to be stated that we do not claim that only 169 earthquakes occurred in that region. Those are events of tectonic origin which haven't been discarded during the initial stage of the detection procedure.*

6) Validation of the results. The authors claim the success of the described methodology. But they do not mention if any test was implemented to validate or to cross-check their results. Common procedure, in seismology, is to visually inspect data to compare automated detection with visual observation. It is mandatory, in my opinion, a validation test, to explore the "efficiency" of the proposed methodology in terms of missed events (of the three kinds) and of false detections. I would be really surprise to see that both numbers (missed and false) are equal to zero. Paper by Kohler et al., mentioned in the Introduction, produces a similar catalogue but this is not discussed in the manuscript. Results are not compared even though I read in Kohler et al. "Most events occurred between July and December, with peak activity in August and September. Seasonal seismicity varies in accordance with expected glacier dynamic activity, . . .."

*Authors: We assume that general assessment of the glacial seismic activity can be achieved basing on the theoretical signal parameters alone. We agree that the robustness and accuracy of the algorithm can be improved, but a visual inspection of calving and relating it to seismically-detectable events is beyond the scope of this study and can be in fact a seed for a separate project.*

*It has to be pointed out that the paper by Kohler et al. (2015) has been published only two weeks before submission of our manuscript. Therefore, we had not enough time to discuss their results sufficiently. Now, the comprehensive discussion, involving detailed comparison of both results, has been included in the revised version of the manuscript.*

7) Seasonality. A description of the variability of the background noise over months (and years) is missing. If noise changes, detection capability of small events changes as well. This test is mandatory prior to explore the seasonality of the number of events.

*Authors: Here we provide plots (Fig. 3) showing background noise variability (daily RMS values) at the HSPB station. RMS was computed using bandpass filtered (1-15 Hz) daily records. It can be inferred that the background noise level does not change significantly from year to year. However, seasonal variations can be observed, having their peaks in the autumn each year. Hence, long-term changes in the observed glacier seismicity are not affected by background noise variability. It can, however, affect seasonal glacier seismicity trends, decreasing the detection capability in autumn evenly each year.*

*For those reasons we resign from including these considerations in the manuscript.*

[Figure]

**Fig. 3 Daily RMS values at the HSPB station (vertical component)**

In seismology, the study of the rates of seismicity over time commonly relies on two concepts: magnitude completeness and declustering . The first prevents the risk of comparing the number of events in two epochs in which the detection threshold was different. The second prevents to include "aftershocks" in the analysis of time-varying rates of events. I wonder if the authors considered these (noise amplitude, completeness, clustering) to prevent a misinterpretation of the variability for the number of events over time.

*Authors: The concept of magnitude completeness is inapplicable for the events without a defined magnitude. Although the nature of icequakes can be compared to the nature of earthquakes, there is no commonly used magnitude definition for icequakes. What is more, we detect icequakes jointly*

*with ice-vibrations which have completely different nature (Górski, 2014). The second procedure - declustering - is used to remove aftershocks, as it was stated by the Reviewer. It is a common procedure in seismic hazard analysis, which has different aims. We do not want to eliminate any glacier-related events from our data. We provide a method to identify them in a bigger set of other detected events (not related to a glacier).*

8) Correlation. The authors claim that glacier-related events originated at Hansbreen and Kronebreen (Page 3, line 4) and later they mention they could not locate them (Page 9, line 4-7).

*Authors: We analysed the same time span of the data as Kohler et al. (2015) but we used single station detections. In case of HSPB, the nearest glacier was Hansbreen and for KBS it was the Kronebreen glacier system. We can assume that our STA/LTA detection algorithm should detect as a minimum the same number of events as it was detected by Kohler et al. (2015), who used the SPITS array located at greater distance than the HSPB, using HSPB records only to verify detection results. In fact, as glacier-generated, we have classified even less events than they did. It indicates that criteria we used were more restrictive than those used by Kohler et al.*

*Hence, we can assume, that our detections include mostly the same events as Kohler et al. (2015) have shown for Hansbreen glacier. And hence, we claim that what we show is a long-term glacier-related seismic activity registered in the vicinity of both Hansbreen and Kronebreen glaciers in Spitsbergen.*

Single station location techniques exist even though they are difficult to implement and these should at least mentioned and discussed. I would remark that a paper was published on this topic (http://link.springer.com/article/10.3103/S0747923915030032) using data from HSPB. This paper is surprisingly not cited.

*Authors: We were unaware of mentioned paper's existence. It was cited now and mentioned techniques were discussed. Thank you for providing this reference.*

Moreover, I remain convinced that a proof about the relation between the observed events and the glaciers' activity is missing. For the case of Greenland, for example, such correlation has been found on the base of further observations as filming or water pressure data.

*Authors: One of the arguments to link those events with glaciers is that they follow the seasonal pattern. Another one is that signals from major sources of non-glacier seismicity has been described by earthquake and false criteria and separated with the help of the fuzzy logic algorithm. Other group of signals have been classified as ice-vibration because their wavelets suits ice-vibrations characteristics. Therefore, the rest is different from typical noise and earthquakes waveforms, but not similar enough to ice-vibrations to be classified so. Glacier however, is also a source of signals different from ice-vibrations, but much harder to specify like e.g. icequakes or different kinds of calving.*

*Because we remove most of non-glacier-induced signals we assume the rest to correspond to glacier activity as supported also by their seasonal distribution.. Those conclusions can be further confirmed by comparing the events' seasonal and interannual distribution with the work of Kohler et al. (2015). As it was stated above (as reply in point 8) "Correlation"), we can assume, that our detections include mostly the same events as Kohler et al. have shown for Hansbreen glacier.*

Any further source of "earthquake like" signal is present in the region? Plants, Mines, Dams and so on?

*Authors: There are only two active mines in Svalbard, located near Longyearbyen, however it can be inferred e.g. from above mentioned paper (Asming and Fedorov, 2015), that no increased seismic activity has been found in the area of those mines.*

Further issues:

- type of used filters is not described (Butterworth?).

*Authors: Yes, we used Butterworth filter. This information was added to manuscript.*

it is the combination of seismometer+digitizer that gives a broadband response.

*Authors: Yes. STS-2 seismometer is a broadband instrument with response from 0.00833 Hz (120s period). It's specifications can be found here:*

*http://www.kinemetrics.com/uploads/PDFs/STS-2.5%20Datasheet.pdf*

- the last quarter of 2007 was included in the analysis but the time-span for the results from HSPB dataset is always referred as (2008-2014). How this affects figure 6a and 6c?

*Authors: The last quarter of 2007 was only used when showing total number of events year to year and was counted to the overall detections number. It was not included when showing seasonal changes. We clarified this issue in the manuscript.*

- As far as I could see, HSPB data at Orfeus data-center start in Jan 2010, am I wrong?

*Authors: It is clarified in the revised manuscript. For the earlier years we copied the data from our in-house storage at Institute of Geophysics, PAS.*

- Page 5, line 5, this selection rule aim to discard "strong tectonic earthquakes", I wonder if the authors refer to those occurring at regional and/or teleseismic distance. This should be specified because in the following of the manuscript they count the detected tectonic earthquakes.

*Authors: We refer to regional and teleseismic earthquakes. Such information has been added to the manuscript.*

- Page 6, line 27, 7020+1858 = 8878 while at page 5 line 11 the number of detected events is 8876

*Authors: We corrected that mistake. Total number of detected events for HSPB of 8876 was correct, while number of detections in the tectonic or false groups of 1858 was wrong. The true value is 1856, which sum up correctly. Thank you for pointing this out.*

- Figures are sometimes references as "Fig." and sometimes as "Figure"

*Authors: This rule can be found in manuscript preparation section on TC website and it states:*

*'The abbreviation "Fig." should be used when it appears in running text and should be followed by a number unless it comes at the beginning of a sentence'*

- Page 7, line 5. I would suggest to first describe the result and then comment on them.

*Authors: Order changed*

- As far as I could see on EIDA server at Orfeus, two further stations exist in the region. Could their data help the discrimination and location of part of the events?

*Authors: We wanted to create an algorithm capable of event recognition with single station use only, therefore we chose to use only HSPB and KBS stations. In the future, similar study can be performed using all available stations which would also extend possible parameters to be used in the classification algorithm.*

- Figure 1, I would suggest to reference the two maps on the left on the right map, to help the reader. Furthermore, I would suggest to write only the relevant toponyms to ease their identification on the map.

*Authors: We've changed the map and we hope that i is more clear now. We tried to use rectangles on a general map of Svalbard, but they were too small in the scale of the map and consequently unreadable. However, we added arrows to make the map more legible.*

*References:*

*Asming V. E. and Fedorov A. V.: Possibility of using a single three-component station automatic detector–locator for detailed seismological observations, Seismic Instruments, 51 (3), 201-208, 2015.*

*Górski M.: Predominant frequencies in the spectrum of ice-vibration events. Acta Geophysica Polonica 52 (4), 657 457–464, 2004.*

*Köhler A., Nuth C., Schweitzer J., Wiedle C. and Gibbons S. J.: Regional passive seismic monitoring reveals dynamic glacier activity on Spitsbergen, Svalbard, Polar Res, 34, 26178, 2015.*

*Sarma S. K.: Energy flux of strong earthquakes, Tectonophysics 11(3), 159-173, 1971.*

*O'Neel S., and Pfeffer W. T.: Source mechanics for monochromatic icequakes produced during iceberg calving at Columbia Glacier, AK, Geophys. Res. Lett., 34, L22502, doi:10.1029/2007GL031370, 2007.*

*O'Neel S., Marshall H. P., McNamara D. E., and Pfeffer W. T.: Seismic detection and analysis of icequakes at Columbia Glacier, Alaska, J. Geophys. Res., 112, F03S23, doi:10.1029/2006JF000595, 2007.*

*Walter F., O'Neel S., McNamara D., Pfeffer W. T., Bassis J. N., and Fricker H. A.: Iceberg calving during transition from grounded to floating ice: Columbia Glacier, Alaska, Geophys. Res. Lett., 37, L15501, doi:10.1029/2010GL043201, 2010.*

---

## Author Comment (AC4) · 17 Apr 2016

AC1 should be removed as it contains wrong supplement. Please refer to AC2